# Derivation of Soil Criteria of Cadmium for Safe Rice Production Applying Soil–Plant Transfer Model and Species Sensitivity Distribution

**DOI:** 10.3390/ijerph19148854

**Published:** 2022-07-21

**Authors:** Xuzhi Li, Junyang Du, Li Sun, Ya Zhang, Yanhong Feng, Liping Zheng, Guoqing Wang, Xinghua Huang

**Affiliations:** 1Key Laboratory of Soil Environmental Management and Pollution Control, Nanjing Institute of Environmental Science, Ministry of Ecology and Environment, Nanjing 210042, China; rmlee40@126.com (X.L.); hjgcdjy@163.com (J.D.); sunli@nies.org (L.S.); zhangya@nies.org (Y.Z.); fyh@nies.org (Y.F.); zlp@nies.org (L.Z.); 2College of Environment Science and Engineering, Yangzhou University, Yangzhou 225127, China; huangxh@163.com

**Keywords:** soil criteria, soil–plant transfer model, species sensitivity distribution, cadmium, rice safety

## Abstract

Widespread soil contamination is hazardous to agricultural products, posing harmful effects on human health through the food chain. In China, Cadmium (Cd) is the primary contaminant in soils and easily accumulates in rice, the main food for the Chinese population. Therefore, it is essential to derive soil criteria to safeguard rice products by assessing Cd intake risk through the soil–grain–human pathway. Based on a 2-year field investigation, a total of 328 soil–rice grain paired samples were collected in China, covering a wide variation in soil Cd concentrations and physicochemical properties. Two probabilistic methods used to derive soil criteria are soil–plant transfer models (SPT), with predictive intervals, and species sensitivity distribution (SSD), composed of soil type-specific bioconcentration factor (BCF, Cd concentration ratio in rice grain to soil). The soil criteria were back-calculated from the Chinese food quality standard. The results suggested that field data with a proper Cd concentration gradient could increase the model accuracy in the soil–plant transfer system. The derived soil criteria based on soil pH were 0.06–0.11, 0.33–0.59, and 1.51–2.82 mg kg^−1^ for protecting 95%, 50% and 5% of the rice safety, respectively. The soil criteria with soil pH further validated the soil as being safe for rice grains.

## 1. Introduction

In recent years, increasing environmental incidents caused by soil contamination due to the intrusion of intensive anthropogenic practices have been reported [1,2]. According to a recent National Investigation Bulletin of Soil Pollution Status in China, about 19.4% of cultivated soils are contaminated [3]. Cadmium (Cd) is the primary contaminant, with 13.7% of soils exceeding the corresponding soil standards in China [3]. Cd is a non-essential element for plants and easily accumulates in crops, such as rice, vegetables, and wheat. These crops are the main source of Cd exposure for the Chinese population, accounting for about 80% of the total Cd intake in China [4]. Epidemiological studies have proven that several Cd-associated diseases in the liver, kidney, and children could increase cancer risk; as a result, it is classified as a human carcinogen (Group 1) [5,6]. Therefore, the widespread soil Cd contamination could impair the quality of agricultural products, thereby threatening human health through the food chain.

The diverse regulatory systems have led to a variety of soil quality standards (SQS) with different protection objectives [7,8,9]. For agricultural soils, an appropriate SQS helps identify the adverse effects of soils on the safety and quality of crop production for protecting human health, which is crucial for agricultural soil risk control and management, as an enforced administrative tool to simplify the initial assessment of environmental risks [10,11,12,13]. Consequently, in the interest of better food safety, the security of rice grains should be a high priority and it is essential to establish the accurate and reliable soil criteria for reducing Cd absorption in rice grains from agricultural soil.

In China, a new risk-based standard, termed the risk control standard for soil contamination of agricultural land (GB 15618-2018), was issued for classification management and the safe use of agricultural soils [14]. The standard proposed risk screening values (RSVs) and risk intervention values (RIVs) to safeguard agricultural products according to different soil conditions (paddy soils and others) and properties (soil pH). For instance, Cd levels exceeding risk values may be present in agricultural products if the Cd content in soils is between the RSVs and RIVs. In contrast, soils contaminated with Cd above the level stipulated by RIVs cannot be restored for cultivation purposes through agronomic regulation and soil remediation [2]. In practice, the Chinese SQS help predict the Cd concentration in edible cultivated products by comparing them with the Chinese food quality standard (FQS) [15]. However, the SQS have been reported to over-protect or under-predict Cd exceedance due to the limited supporting data [16,17,18,19]. Therefore, studying soil criteria is needed to effectively ensure food safety and revise the SQS in the future.

The key to deriving soil criteria for Cd accumulation is to develop a reliable and reasonable prediction model describing soil–plant transfer (SPT). The model should fully consider the bioavailability factors (such as soil pH, organic matter, and clay content) and the cultivar sensitivities [20,21,22,23]. However, the published prediction models, such as the Freundlich-type equation, were mainly derived from the metal-spiked soils in pot and greenhouse experiments [17,22,24]. Such transfer models fail to extrapolate the realistic conditions due to the bioavailability-relevant discrepancies between the field and experimental studies [19,23]. Moreover, poor relationships between Cd levels in field soils and crops have been observed, likely resulting from the sampling objective for the pollution survey rather than the soil criteria study [25]. Thus, it was hypothesized that the criteria-designed field sampling that covers the Cd concentration gradient for both soils and crops could improve the performance of SPT models [19].

Simple generic thresholds derived in previous studies were more likely to cause uncertainty which might fail to identify the risk of Cd accumulation in crops [13,19]. Recently, a probabilistic method based on the SPT model assessed the impact of model uncertainty, calculating the soil thresholds with different probabilities (protective levels) by introducing prediction intervals [26,27,28]. In addition, another probabilistic method is species sensitivity distribution (SSD), which is widely used in deriving SQS and risk assessments [9,29]. The distribution-based approach utilizes species sensitivities, soil types, properties, and pollution sources, with relatively low uncertainty to estimate soil thresholds [29]. Furthermore, SSD for agricultural product safety is modeled based on the bioconcentration factor (BCF) of the soil–crop paired sites rather than the ecotoxicological data of the species. This approach back-calculates the soil thresholds from the FQS with different protective levels [30].

The current study conducted criteria-designed field surveys of soil–rice grain paired samples. The physical and chemical properties of the samples along with the Cd concentrations were assessed. The specific objectives of our research are: (1) to characterize Cd transfer to rice grain and qualify the main influencing factors; (2) to develop reliable SPT and SSD models that would derive soil probabilistic criteria for Cd; (3) to validate the soil criteria concerning rice safety for Cd accumulation.

## 2. Materials and Methods

### 2.1. Study Sites and Sampling

The study covers China’s major rice-producing areas, including ten provinces (Anhui, Fujian, Guangdong, Guangxi, Hubei, Hunan, Jiangsu, Jiangxi, Liaoning, and Zhejiang). The sampling sites were designed to collect the paired samples with wide Cd concentration ranges, soil types, rice cultivars and physicochemical properties based on the distance (10–1000 m) from the pollution sources (such as metal mining sites, industrial sites, and other contaminated areas) (Figure 1).

At harvest time between 2014 and 2016, 328 pairs of surface soil and rice grain samples were collected (Figure 1). Approximately 1.0 kg of the surface soils (0–20 cm), consisting of five sub-samples, were sampled using a grid method, and 0.5 kg of fresh samples of rice grains was collected at each site where the soils were collected. The growth conditions were also recorded. The mixed soil samples and fresh edible parts of the rice were stored in clean plastic bags. At the lab, soil samples were dried and ground to pass through 2 mm and 0.1 mm sieves before chemical analyses. Rice samples were rinsed with ultrapure deionized water, oven-dried at 60 °C for 48 h, and ground using a ball mill before chemical analyses.

### 2.2. Chemical Analyses

Both field and experimental studies demonstrated that the Cd uptake of rice grains were significantly related to the soil Cd level, soil pH, organic matter content (SOM), and clay content in China [6,16,19]. Thus, the soil Cd concentration, pH, SOM, and clay content were selected to construct the SPT model in our study. 

The soil pH (soil:water = 1:2.5) was determined after shaking the soil sample in distilled water for 30 min. The soil organic carbon content was determined using an Elemental Analyzer after the HCl treatment method for 24 h. The SOM was obtained by multiplying the values of organic carbon by 1.724. Soil clay contents were measured using laser diffraction [31].

For Cd determination, 0.25 g of air-dried soils was digested with a mixture of 10 mL HCl, 5 mL HNO_3_, 3 mL HF, and 3 mL HClO_4_. The digested extracts were then dissolved with 2 mL HCl (1:1, *v*/*v*) and diluted to 50 mL with deionized water for the Cd content analysis [32]. Microwave-assisted digestion of rice samples was conducted using a HNO_3_-HClO_4_ mixture [33,34]. The standard reference materials used were as follows: GSS-5 and GSB-8 were also used during the digestion process for quality assurance and control of soils and rice grains, respectively. Inductively Coupled Plasma Mass Spectroscopy was used to measure the Cd concentration [33,34]. The values are presented based on the dry weight (DW) for soils and rice grains.

### 2.3. Data Analysis

#### 2.3.1. Constructing the Soil–Plant Transfer Model

The relationship between Cd concentrations in soils and rice grains was established using the empirical Freundlich type equation [24]:Log[Cd_rice_] = a × pH + b × Log[Cd_soil_] + c × Log[SOM or Clay] + k(1)
where Cd_soil_ and Cd_rice_ refer to Cd concentrations (mg kg^−1^) in soils and rice grains, respectively. SOM and Clay indicate soil organic matter (g kg^−1^) and clay content (%), respectively. a, b, c, and k are the fitted parameters derived by stepwise linear regression.

Then, the SPT model was used to predict the Cd concentration in soils according to the given values for soil properties (pH, SOM, or clay) and the Cd content in rice grains. To calculate the probabilistic soil thresholds based on a series of protection levels, the SPT model together with one-sided prediction intervals were introduced, and soil thresholds were determined with the limits of these prediction intervals equal the Chinese FQS (0.2 mg kg^−1^ for rice grains) [26,27,28].

#### 2.3.2. Constructing the Species Sensitivity Distribution

The BCF, representing the transfer characteristic of soil–rice grain paired samples (Equation (2)), was used to construct the SSD model [16].
(2)$$\mathrm{BCF}=\frac{{\mathrm{Cd}}_{\mathrm{rice}}}{{\mathrm{Cd}}_{\mathrm{soil}}}$$
where BCF is the bioconcentration factor.

The distributions of Burr III, Gamma, Log-Gumbel, Log-logistic, Log-normal, and Weibull were applied to fit the SSD curves of the 1/BCF data according to the various soil conditions, such as soil pH [35,36]. The goodness-of-fit analysis (Anderson–Darling (AD) statistic, Kolmogorov–Smirnov (KS) statistic, Akaike’s Information Criterion (AIC), Bayesian Information Criterion (BIC) and Akaike’s Information Criterion corrected for sample size (AICc)) was conducted to evaluate the fitting performance of SSD models and select the best distribution model. Generally, the best distribution model was the SSD with an AICc parameter (delta) of 0.

Furthermore, the hazardous BCFs were derived based on a series of protection levels in SSD curves. Then, the corresponding soil thresholds (hazardous concentrations, HC_x_) for Cd were back-calculated from these BCFs and the corresponding FQS [17].

#### 2.3.3. Deriving and Validating the Soil Criteria

In this study, the probabilistic soil criteria for protecting the 95% (SC_5_), 50% (SC_50_), and 5% (SC_95_) rice safety were determined as the minimum values of thresholds derived by the SPT models and SSDs. SC_5_ and SC_95_ referred to the Chinese RSVs (slight risk) and RIVs (severe risk), respectively [14], while SC_50_ was used to indicate the moderate risk [27]. 

The Chinese agricultural soil standards provide information on whether soils can produce safe crops [14]. Thus, the quality of rice production was used as the validation basis of soil criteria in our study. First, SC_5_, corresponding to the RSV_S_, and SC_50_, indicating the controllable risk, were evaluated according to the proposed suitability classification method [26]. As shown in Table 1, Categories A (unsuitable) and C (suitable) indicated that soil criteria predict the soil quality for proper rice production. In contrast, Categories B (false negative) and D (false positive) referred to incorrect predictions of the production quality of agricultural soils. For each paired sample, the Cd concentrations in the soil and paired rice samples were compared with the soil criteria and the Chinese FQS of rice grains.

Secondly, SC_95_ validation corresponding to the RIVs was conducted according to the safety assessment of the paired rice grains [14]. The exceeding factor (E) was calculated as the ratio of Cd concentration in rice grains to the corresponding FQS, and divided into three categories including safe (E ≤ 1), moderate (1 < E ≤ 2), and severe (E > 2). The percentage of rice samples with severe type was the main factor used to evaluate these criteria.

#### 2.3.4. Statistics Analysis

Spearman’s correlation analysis was performed to identify the relationships between the Cd concentration in soils and rice grains, as well as soil properties (soil pH, SOM, and clay content) (SPSS 26.0, IBM, Armonk, NY, USA). The SPT model was constructed using stepwise linear regression and the probabilistic soil thresholds, depending on the one-sided prediction intervals, were calculated using JMP 16.0 software (SAS, Cary, NC, USA) The polynomial surface fit analysis was conducted using Origin 2018. The model analyses of SSD curves, the goodness-of-fit analysis, and the calculation of probabilistic HC_x_ were conducted based on 1/BCF using the ssdtools package in R3.3.6. 

## 3. Results

### 3.1. Cd Concentrations in Soils and Crops

As shown in Table 2, the mean soil Cd concentration was 1.19 (0.007–17.9) mg kg^−1^ for the 328 soil–rice samples. Among the rice grains for the field surveys, the Cd concentration varied from 0.003 to 4.87 mg kg^−1^, averaging 0.550 mg kg^−1^. The median values of Cd concentration in soils and paired rice grains were 0.563 and 0.202 mg kg^−1^, respectively. The BCFs of rice grain were of approximately five orders of magnitude, from 0.004 to 10.1. The mean and median values of BCFs were 0.730 and 0.393, respectively.

The soil samples covered a wide variation of physico-chemical properties. Soil pH values ranged from highly acidic (pH ≤ 5.5), acidic (5.5 < pH ≤6.5), neutral (6.5 < pH ≤ 7.5) to alkaline (pH > 7.5), with a mean of 5.74. The mean SOM was 3.63 g kg^−1^, from 1.24 to 8.96 g kg^−1^, while the average clay content of soil samples was 28.2 (6.60–61.8) %.

The soil Cd content ranged from background levels to heavily polluted soils. Compared to the Chinese soil standards, 64.3% (211 out of 328) of soil samples were above the RSVs, while 19.2% (63 out of 328) of soils had a Cd concentration exceeding the RIVs (Figure 2a). The exceeding sites were concentrated in highly acidic and acidic soils. With regard to the rice Cd content, 167 rice samples (50.9%) had values exceeding the maximum limits, namely the Chinese FQS (Figure 2b). 

### 3.2. Soil–Plant Transfer Models for Cd Accumulation

Figure 3 summarizes the relationships between Cd accumulation in the soil–rice system and soil properties. In the present study, the soil Cd concentration predominantly affected the Cd accumulation of rice grains, with Spearman’s correlation coefficients of 0.431 (*p* < 0.05). Soil pH correlated negatively with Cd concentration in rice grains significantly (*R* = −0.328, *p* < 0.05), while soil SOM and clay content had no effects on Cd accumulating in grains. Furthermore, soil pH and SOM had significantly negative relationships with the BCFs, with coefficients of 0.281 (*p* < 0.05) and 0.212 (*p* < 0.05), respectively. 

Stepwise multiple linear regression was performed to construct the soil properties-dependent Freundlich-type function. In this study, soil pH-dependent equations were obtained for soil–rice paired samples as follows:Log[Cd_rice_] = 0.535 × Log[Cd_soil_] − 0.187 × pH + 0.474 (*R*^2^ = 0.266, *p* < 0.001)(3)
Log[Cd_rice_] = 0.857 × Log[Cd_soil_] − 0.074 × pH + 0.083 (*R*^2^ = 0.638, *p* < 0.001)(4)

Compared to the poor predictive ability of the model based on all the soil–rice samples, removing the outliers located outside the whiskers of BCF boxplots of rice grains could increase the model accuracy to forecast the Cd concentration in rice grains, expressed by the regression coefficient from 0.266 (n = 328) (Equation (3)) to 0.638 (n = 267) (Equation (4)). In addition to linear regression, polynomial surface models were also formulated due to the complex Cd transportation mechanism in the soil–crop system. However, the regression coefficient indicated that no apparent enhancement of the fitting precision was found compared to the Freundlich-type function (Figure 4). 

The applicability and accuracy of the SPT prediction model were determined by plotting the measured Cd concentration in rice grains against the corresponding predicted Cd concentration (Figure 5). A significantly linear relationship (*R*^2^ = 0.639, *p* < 0.001) was found between the measured and predicted Cd concentrations of rice grains, and most rice samples were located within the lines of 95% prediction intervals. Thus, Equation (4) could provide reliable predictability of Cd transfer from the sampled soils to the rice grains.

### 3.3. Species Sensitivity Distribution for Cd Accumulation

As mentioned above, soil pH is the critical factor affecting Cd uptake in the soil–rice grain system. Thus, the model analyses were performed to construct the SSD curves based on different soil pH ranges. The BCF dataset was divided into the following four groups: highly acidic (pH ≤ 5.5), acidic (5.5 < pH ≤ 6.5), neutral (6.5 < pH ≤ 7.5) and alkaline (pH > 7.5) according to the soil pH ranges of GB 15618-2018 standard in China [14]. 

As shown in Figure 6, six SSD models (Burr III, Gamma, Log-Gumbel, Log-logistic, log-normal, and Weibull) were fitted to the BCF dataset of soil pH groups, respectively. Results of fitting the distributions are presented in Table 3. Both the SSD plots and goodness-of-fit statistics (AD and KS statistics) showed that all the SSD models had a good performance in fitting the BCF data, exhibited by AD and KS parameters of more than 0.05; however, none of these distributions ranked the highest for both goodness of fit statistics. Thus, the best distribution models using the AICc parameter (delta) were Burr III (highly acidic soil) and Log-normal (acidic, neutral, and alkaline soils), respectively. 

### 3.4. Derivation and Validation of Soil Criteria

Soil probabilistic criteria (SC_5_, SC_50_, and SC_95_) for rice safety at the four scenarios (according to soil pH ranges) were the minimum thresholds calculated from the SPT models (Appendix A) and SSDs (Appendix A). As shown in Table 4, the final derived SC_5_, SC_50_, and SC_95_ were 0.06–0.11, 0.33–0.59, and 1.51–2.82 mg kg^−1^, respectively. In comparison with Chinese soil standards of the paddy soil, the extremely low values of SC_5_ were lower than the corresponding RSVs (0.30–0.80 mg kg^−1^), even more so than the background levels in China [37]. Contrarily, the SC_50_ values (0.33 and 0.40 mg kg^−1^) were similar to the RSVs (0.30 and 0.40 mg kg^−1^) at soil pH ≤ 6.5, while more rigorous SC_50_ values (0.49 and 0.59 mg kg^−1^) were found compared to the RSVs (0.60 and 0.80 mg kg^−1^) at soil pH > 6.5. Similarly, the highly acidic and acidic soil’s SC_95_ values (1.51 and 1.85 mg kg^−1^) were comparable to the RIVs (1.50 and 2.00 mg kg^−1^), while these criteria (2.27 and 2.82 mg kg^−1^) were obviously less than the corresponding RIVs (3.00 and 4.00 mg kg^−1^) in neutral and alkaline soils. 

The derived soil criteria were then validated based on the suitability classification. A false-negative scenario (category B) was particularly undesirable because the rice was predicted to grow safely, whereas the measured Cd concentration in crops exceeded the corresponding FQS. Moreover, category D representing over-protection should also be emphasized. In this study, due to the extremely low limits of SC_5_, only SC_50_ was selectively evaluated based on the investigated samples (Figure 7a). The misjudged proportions of the rice safety for category B were less than 10% in different soil pH groups, while 24 (15.1%), 20 (27.0%), 15 (32.6%), and 10 (26.3%) rice samples were classified as category D, respectively. The results showed that the controllable criteria, SC_50_, performed well at the primary screening stage considering the safety of rice production. 

Soils unsuitable for cultivating crops were indicated by conducting the safety assessment of rice products (Figure 7b). According to the analysis of the field survey data, 90% of rice grains had a Cd concentration exceeding the FQS, when the paired soil Cd content was above the SC_95_. Furthermore, 54% of these samples were classified as the severe type, which was almost 4 times that of the controllable thresholds SC_50_. These results indicated a high probability that the quality of rice production would not be ensured with the current agronomic technical measures when the soil Cd concentration is above the SC_95_ limit. 

## 4. Discussion

In China, the safety of crop production is one of the key aspects of soil pollution prevention and control in agricultural land. Therefore, in order to protect crop quality with respect to soil contamination, it is crucial to understand the transfer of pollutants from soils to crops, especially for heavy metals [26]. In this study, a designed field sampling of the paired soil–rice sites was conducted, with the Cd concentrations from the background to the heavily contaminated levels. The BCFs collected in our study were much higher than the published BCFs of field data from China in a previous study [38]. Moreover, the ranges of BCFs were about one order greater than the ranges of BCFs in other countries [9,39,40]. This might be due to the larger range of Cd concentrations in soils and rice grains and the diversity of soil types, contamination sources and rice cultivars obtained in this study. 

After establishing the relationship between soils and crops, soil criteria for safeguarding crop products could be reversely derived based on the transfer model and FQS. Previous studies established several models for Cd phytoavailability in soil–plant systems, including mechanism models, such as the free ion activity model (FIAM), the biotic ligand model (BLM) [41,42], empirical models, and SPT models [38]. However, difficulties in obtaining parameters and differences between soil and soil solution limit the application of mechanism models [43]. In practice, empirical models are more applicable with easier predictions and a more detailed application scope, and the relationship between metal concentrations in soils and paired crops could be easily described using the regression equation [21,22]. In SPT models, it has been well recognized that the spiking of metal-salts in pot and greenhouse experiments probably over or under predict the realistic phytoavailability compared to field sites. This was probably due to the leaching and aging of heavy metals in field contaminated soils [44]. Our results revealed that the soil type-specific data generated from the field condition with a realistic Cd concentration gradient could minimize the uncertainty caused by metal-spiked tests and soil heterogeneity, increasing the model performance in the soil–plant transfer system. Thus, it is essential to derive the soil criteria based on the field dataset. 

It was established that pH is the main soil property affecting the bioavailability of heavy metals in soils and plants [45], and increasing soil pH results in decreased Cd mobility and potential availability via co-precipitation with Ca^2+^ and Mg^2+^ [46]. In most cases, soil pH improves the Cd prediction in crops, compared with the single soil–Cd-based model, and should be accounted for when deriving soil thresholds. However, previous studies proposed that it is not sufficient to consider only soil pH to derive the soil thresholds, and other relevant soil properties should be included [47]. Nevertheless, our findings revealed that combining the total soil Cd concentration with pH performed excellently to predict the exceeding risk of agricultural products. Soil pH was the only key factor affecting Cd uptake in this study, so other soil properties were excluded in our SPT model. This might be explained by covariance phenomena, and many studies have demonstrated that soil pH could affect other soil properties, which in turn influence metal uptake in crops [48]. 

Several studies reported contradictory phenomena such as the inconsistent exceedance of soil samples and the paired crops when using soil standards [19,49,50]. In fact, as a country with a vast territory and significant soil heterogeneity, numerous factors (such as parent materials, soil physical-chemical properties, topographic factors, hydrological factors, etc.) could affect human health through Cd accumulation in edible crop parts [2]. Moreover, human activities, such as atmospheric pollution and sewage irrigation, also contribute to the exceeding risk posed to agricultural products. Thus, the exceedance of agricultural commodities is a probability regarding the soil standards, and the occurrence of unrelated small probability events, such as Categories B and D, could not be entirely excluded due to the unavoidable uncertainties. Therefore, two probabilistic methods were derived in this study to reduce the model uncertainty, including the SPT model with one-sided prediction intervals, and the SSD composed of soil type-specific BCF. 

The SPT model expressed as a regression equation offers the possibility of calculating probabilistic soil criteria from food safety limits. Given the FQS and a chosen protective level of exceeding this standard, the Cd thresholds in soils can be calculated by the inverse use of the regression model and taking the prediction intervals into account [28]. In Germany, the government used SPT models with prediction intervals to determine the trigger values and action values of heavy metals and organic contaminants in agricultural soils [51]. In our study, the transfer of Cd from soils to rice grains was influenced by soil pH, thus the probabilistic thresholds in soils were calculated depending on the different soil pH ranges. 

The variety of the rice cultivars is also an important factor affecting Cd accumulation. Thus, the SSD method taking the diversity of rice cultivars and soil properties into account was used to model the BCF data to derive the soil thresholds [1]. Combining the SSD and SPT model as a normalization tool for BCF in metal-spiked tests was performed to derive the soil thresholds for food safety in previous experimental studies [1,52]. However, its normalization has been debated since its inception, because it involves a new uncertainty of the normalization process and the potentially unreliable applicability of normalized equations [44]. Thus, our study does not recommend normalizing the BCFs representing the realistic accumulating status of crops from field sampling. Moreover, selecting the best distribution model is the key to deriving reliable thresholds when using the SSD method. However, there is no theoretical basis and consensus about the precedence of distribution [53]. Several studies have recommended that Burr III distribution often presents the best fit of SSD curves [53], which did not agree with our finding of the Log-normal distribution suitable for most scenarios. This occurrence might be explained by the BCF data from field surveys being adequate to reflect the actual rice sensitivities for Cd bioaccumulation. 

In most cases, the SQSs among different countries could be classified into different risk categories, broadly termed negligible (slight) risk as long-term quality objectives, warning (moderate) risk as the trigger values and potentially unacceptable (severe) risk as remediation needed [49]. Thus, the traditional model to calculate a genetic threshold was not applicable to derive the SQS with a series of risk levels. The proposed probabilistic approach combining the SPT model and SSD extrapolation will assist in determining SQS for different land use purposes with different levels of risk. However, the final determination from SPT and SSD were not clear in practice. For convenience, the final soil criteria were the minimum thresholds based on the SPT model and the SSD curves according to the conservative principle in this study. The three sets (SC_5_, SC_50_, and SC_95_) of criteria were comparable to the Chinese SQS in highly acidic and acidic soils, while stricter soil criteria were observed in neutral and alkaline groups. Compared to the previous studies from field data, the probabilistic criteria were stricter, especially in alkaline soil conditions [38]. Compared to the other countries, the proposed criteria were also rigorous in our study. The agricultural soil Cd standards set by the United Kingdom (1.80 mg kg^−1^), Brazil (3.00 mg kg^−1^), Belgium (2.00–10.0 mg kg^−1^), Canada (1.40 mg kg^−1^), Austria (1.00 mg kg^−1^), the Netherlands (0.60 mg kg^−1^), Korea (4.00–12.0 mg kg^−1^), and Thailand (37.0 mg kg^−1^) pertain to all soil types [7,16,49]. SQSs for the Czech Republic, Slovakia, and Germany are set as 0.4–1.0, 0.4–1.0, 0.4–1.5 mg kg^−1^ according to soil texture [16,49]. The SQSs of Cd set by the European Union are 0.5, 1.0, and 1.5 mg kg^−1^ for soils according to soil pH, respectively [16]. Our results revealed that the derived soil criteria were strict enough, and the controllable criteria, SC_50_, can be used safely to produce safe rice, while more rigorous thresholds such as SC_95_ could indicate severe soil contamination regarding the risks of Cd exceedance in food crops.

## 5. Conclusions

Using rice grain as an example, this study derived soil criteria for Cd pollution based on food quality standards by applying th following two probability-based approaches: SPT models and SSDs. Our results suggest the following:The field data with a proper Cd concentration gradient could increase the model accuracy in the soil–plant transfer system and decrease inaccuracy in the derivation of soil criteria.The probabilistic soil criteria (SC_5_, SC_50_ and SC_95_) for protecting 95%, 50% and 5% of the rice safety were 0.06–0.11, 0.33–0.59, and 1.51–2.82 mg kg^−1^ according to soil pH ranges, respectively.The proposed soil criteria were comparable to the Chinese SQS in highly acidic and acidic soils, while more strict soil criteria were observed in neutral and alkaline groups.

## Figures and Tables

**Figure 1 ijerph-19-08854-f001:**
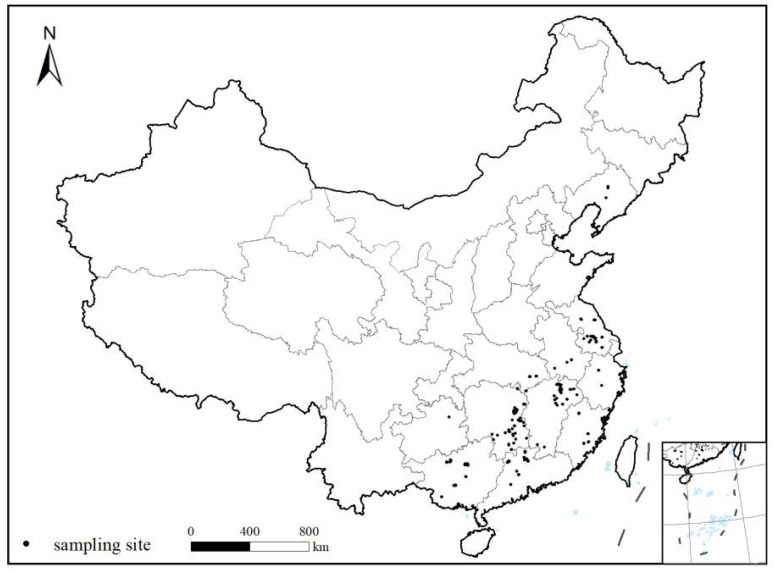
Spatial distribution of soil–rice grain sampling sites in China.

**Figure 2 ijerph-19-08854-f002:**
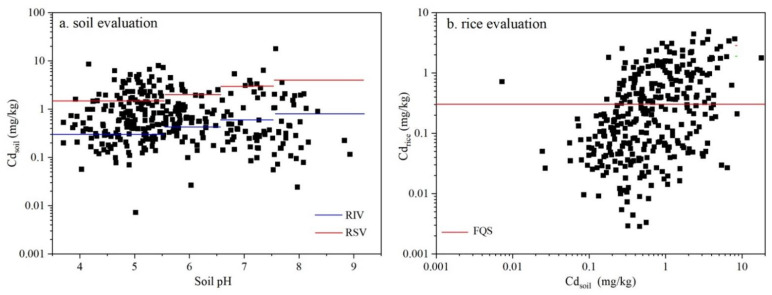
Evaluation of soil–rice grain paired samples based on the Chinese soil and food standards.

**Figure 3 ijerph-19-08854-f003:**
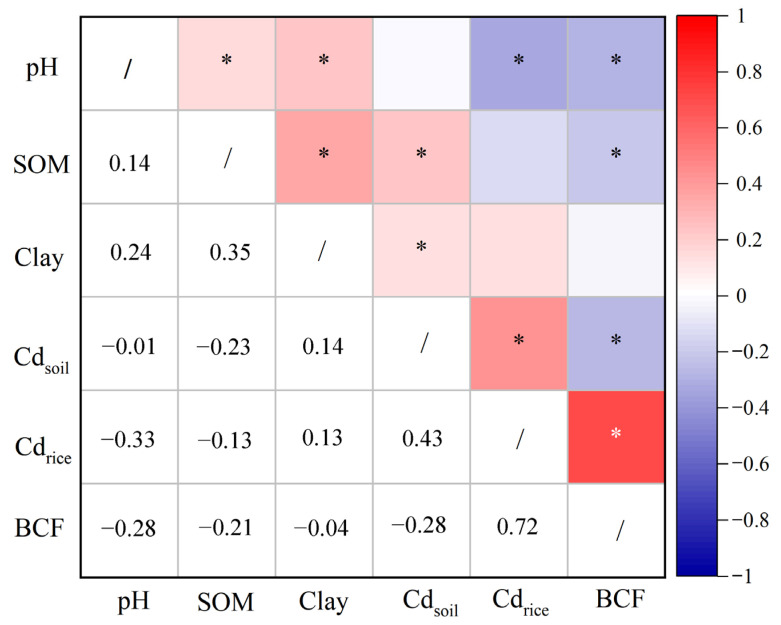
Correlations between Cd concentrations in soils and rice grains as well as soil properties. The color of red and blue indicate positive and negative relationships, respectively. * denotes significant correlation at *p* < 0.05.

**Figure 4 ijerph-19-08854-f004:**
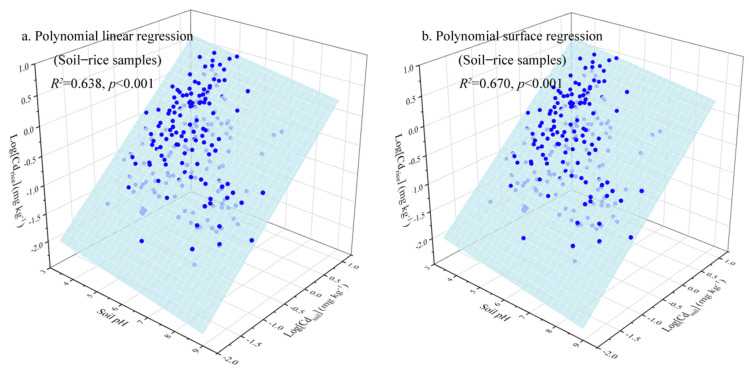
Relationships between Cd uptake by rice grains, soil Cd concentrations, and soil pH.

**Figure 5 ijerph-19-08854-f005:**
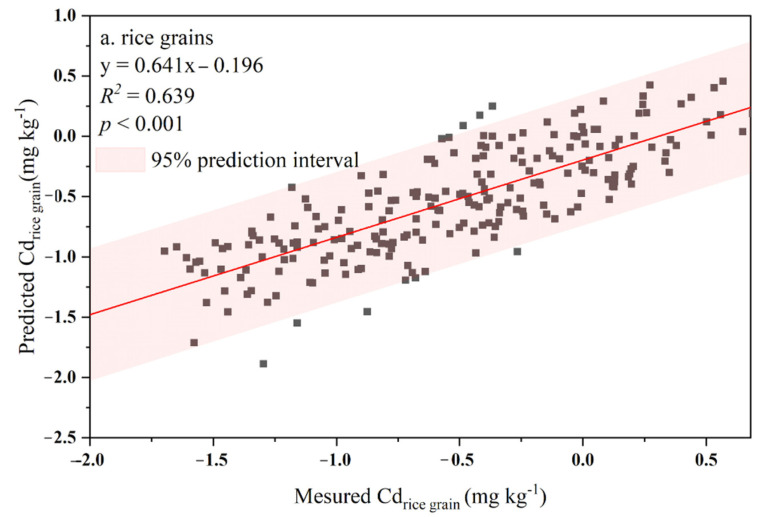
Relationships between the measured and predicted concentrations of Cd in rice grains. The red area represents the 95% prediction interval.

**Figure 6 ijerph-19-08854-f006:**
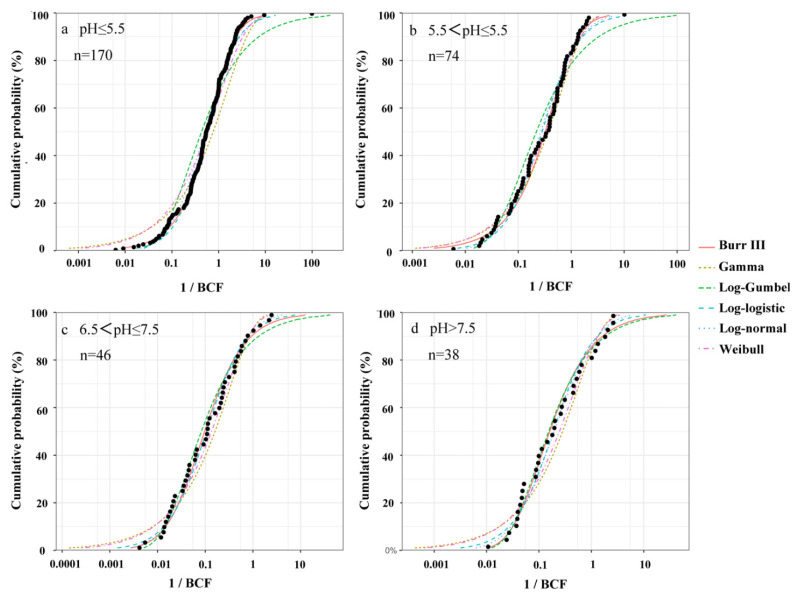
Species sensitivity distribution of rice grains for accumulating Cd according to different soil pH ranges.

**Figure 7 ijerph-19-08854-f007:**
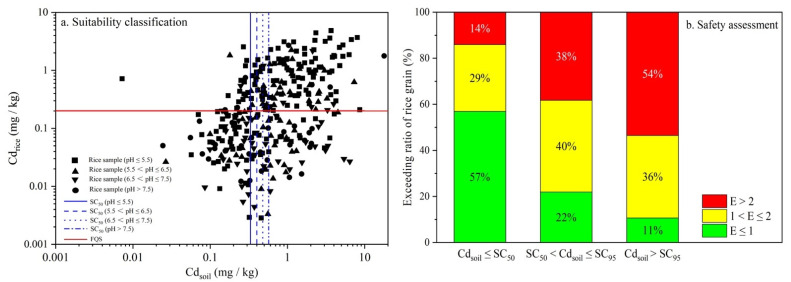
Suitability classification and safety assessment of rice grain samples based on SC_50_ and SC_95_.

**Table 1 ijerph-19-08854-t001:** Suitability classification of the soil–rice paired samples.

Suitability Classification	Soil	Rice Grain
A	Cd_soil_ > SC_x_	Cd_rice_ > FQS
B	Cd_soil_ ≤ SC_x_	Cd_rice_ > FQS
C	Cd_soil_ ≤ SC_x_	Cd_rice_ ≤ FQS
D	Cd_soil_ > SC_x_	Cd_rice_ ≤ FQS

Cd_soil_ and Cd_rice_ refer to the Cd concentration in soil and paired rice grain, respectively. SC_x_ and FQS refer to the derived soil criteria and Chinese food quality standard, respectively.

**Table 2 ijerph-19-08854-t002:** Descriptive statistics on Cd concentrations in soils and rice grains as well as soil properties.

Variables	Min	Max	Mean	Median	SD
Cd_soil_ (mg kg^−1^)	0.007	17.9	1.19	0.563	1.65
Cd_rice_ (mg kg^−1^)	0.003	4.87	0.550	0.202	0.783
BCF	0.004	10.1	0.730	0.393	5.48
pH	3.70	8.93	5.74	5.45	1.14
SOM (g kg^−1^)	1.24	8.96	3.63	3.58	19.8
Clay (%)	6.60	61.8	28.2	28.2	10.7

**Table 3 ijerph-19-08854-t003:** Goodness-of-fit of distribution models for soil–rice samples.

pH	Models	AD	KS	AIC	BIC	AICc (Delta)
pH ≤ 5.5	**Burr III**	**0.290**	**0.037**	**346**	**356**	**0**
	Gamma	/	0.150	437	444	90.9
	Log-Gumbel	5.87	0.129	406	412	59.5
	Log-logistic	0.816	0.053	350	356	3.25
	Log-normal	1.23	0.069	355	362	8.92
	Weibull	3.49	0.098	392	398	45.1
5.5 < pH ≤ 6.5	Burr III	0.424	0.080	67.8	74.7	2.24
	Gamma	0.770	0.080	75.1	79.7	9.36
	Log-Gumbel	1.83	0.120	82.5	87.1	16.7
	Log-logistic	0.652	0.088	67.9	72.4	2.15
	**Log-normal**	**0.602**	**0.100**	**65.7**	**70.3**	**0**
	Weibull	0.477	0.060	70.4	74.9	4.63
6.5 < pH ≤ 7.5	Burr III	0.356	0.086	25.7	20.1	4.87
	Gamma	0.956	0.125	22.2	18.5	8.02
	Log-Gumbel	0.509	0.094	26.0	22.3	4.23
	Log-logistic	0.345	0.076	7.31	23.6	2.94
	**Log-normal**	**0.260**	**0.072**	**30.3**	**26.5**	**0**
	Weibull	0.514	0.087	25.5	21.8	4.69
pH > 7.5	Burr III	0.350	0.098	18.8	23.4	3.41
	Gamma	1.02	0.139	21.7	24.7	5.86
	Log-Gumbel	0.347	0.089	17.1	20.1	1.26
	Log-logistic	0.427	0.104	18.4	21.4	2.56
	**Log-normal**	**0.396**	**0.108**	**15.8**	**18.9**	**0**
	Weibull	0.712	0.112	19.9	22.9	4.09

AD, KS, AIC, BIC, AICc refer to Anderson Darling, Kolmogorov-Smirnov, Akaike Information Criterion, Bayesian Information Criterion and Akaike Information Criterion corrected for sample size (indicated by parameter delta). The best fitting models are presented in bold.

**Table 4 ijerph-19-08854-t004:** The derived soil criteria (SC_5_, SC_50,_ and SC_95_) of Cd for rice safety according to soil pH ranges.

Soil Criteria (mg kg^−1^)	pH ≤ 5.5	5.5 < pH ≤ 6.5	6.5 < pH ≤ 7.5	pH > 7.5
SC_5_	0.06	0.08	0.11	0.09
SC_50_	0.33	0.40	0.49	0.59
SC_95_	1.51	1.85	2.27	2.82

SC_5_, SC_50_, and SC_95_ refer to soil criteria indicating the slight, moderate, and severe risk, respectively.

## Data Availability

Data are available within the article or Appendix A.

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
