# Peer review of "Derivation of Soil Criteria of Cadmium for Safe Rice Production Applying Soil–Plant Transfer Model and Species Sensitivity Distribution"

_ijerph, 2022, doi:10.3390/ijerph19148854_

Round 1

Reviewer 1 Report

This work investigated the relationship between soil properties and rice grain Cd concentrations and recommended the threshold soil Cd concentrations for safe rice grain production. It is an interesting work, carried out during the hard time when some of the rice grains in China still have unsafe levels of Cd.

My suggestions include,

i)How about dividing your samples into four or more groups according to soil pH and doing the regression? You may find different parameters for the equations, and this may realize more accurate prediction of rice grain Cd concentrations.

ii)The choosing of these soil properties should be discussed, why did not you include CEC?

iii)The potential effect of rice varieties should also be discussed, although the big sample number may compensate this problem to a certain extent.

iv)The discussion section should be re-written, it discussed too much of references, but what is the relationship of results with your findings? And what it the significance and reliability of your results?

iv)”respectively” should be used in Line 124; how did you determine the clay contents? Line 361, I do not think pesticide abuse is serious in China; Line 458, the standard has two authors; Line 488, L should not be italic; Line 490, this book was published in 2000.

v)Collecting wheat samples should be done at the sites where the soils were collected, this should be described with more details so a reliable correlation can be got between soil and wheat Cd.

Reviewer 2 Report

The manuscript number ID: ijerph-1803652 was developed in order to derive soil criteria to safeguard agricultural products by assessing Cd intake risk through the soil-grain-human pathway. Based on a 2-year field investigation, a total of 328 soil-rice grain paired samples were collected in China, covering a wide variation in soil Cd concentrations and physicochemical properties. Two probabilistic methods used to derive soil criteria are soil-plant transfer models (SPT), with predictive intervals, and species sensitivity distribution (SSD), composed of soil type-specific bioconcentration factor (BCF, Cd concentration ratio in rice grain to soil). The soil criteria were back-calculated from the Chinese food quality standard.

The paper contains original results and the overall presentation is convincing.

The following minor corrections are recommended before paper publication:

1. Please refer in more detail and in clearly way the advantages of the methodology and summarize possible limitations.

2. Please enrich the Introduction with a paragraph explaining the correlation between the following sections. 

Reviewer 3 Report

The study concerns a well-developed issue, although still important and current, of the transfer of heavy metals (here Cd) from the soil to the plant. The paper presents a slightly different and interesting approach to this problem. In general, the manuscript was written correctly and in accordance with all standards, but the authors did not avoid some imperfections, which absolutely should be corrected. Below my suggestions and comments were given that should be completed in the new version of the paper:
Abstract
1st line 12 - Rewrite to make it clear contamination is not just a Cd problem. Later it can be specified that the problem raised in a given area concerns only this metal.
2. line 22 - the same sentence from the conclusions - inadmissible, please rewrite it
3. line 26 - this sentence should be deleted, because it cannot be stated on the basis of these studies, where only Cd was analyzed. Each metal behaves differently and undergoes to different influences.
Introduction
1. The objectives of study should be redrafted in such a way as to make it clear that the authors only refer to Cd.
 Material and Methods
1. line 98 - 100 - the distance from the emission sources should be completed.
2.complete the soil systematics and add the soil texture - it is a very important factor in the analysis of heavy metals
3. line 113 -114 - is there no error here? since when do we mark TOC in HCl ??? it is not necessary to use the given apparatus and, moreover, HCl is not suitable for the determination of TOC
4. line 117 - instead of this meaningless vague, please give a specific method. Please remember that the routine analysis is different for each country and laboratory.
5. line 132 onwards - please change the wording from mg / kg to mgkg-1
6.line 139 - in order for the paper to be readable for an international group of recipients, also other than Chinese standards should be provided, e.g. European or according to FAO
7. lines 147 - 153 - edit, because the text is illegible, unclear
8. line 158 - 160 - why such ranges were used - please explain? what were they guided by? if it is according to a standard, please quote it and explain it as well.
Results
1. CEC is missing in table 2 and data interpretation - should be completed. As I mentioned earlier in the case of soil texture, also CEC is an important factor in the sorption processes of heavy metals, and thus its potential bioavailability and absorption by plants.
Figures 3, 4, 6, 7 - their quality should be improved, in their present form they are not valid
line 247 - and others - please give a quotation - should it be [14]?
Discussion
line 334 - 345- please remove the repeating fragments
Conclusions
conclusion 3 - is unfounded (it was previously stated in the abstract) - it cannot be stated so when only one metal has been analyzed. This does not warrant such generalizations

Reviewer 4 Report

Lines 145-153 should be checked for technical errors

Discussion should be improved. Please compare your findings with similar where both SPT and SSD were used in order to determine the fate of Cd in natural ecosystems.

Round 2

Reviewer 3 Report

The manuscript has been significantly revised and the authors will address most of the comments by including them in new version